

# Short-chain soluble polyphosphate fertilizers increased soil P availability and mobility by reducing P fixation in two contrasting calcareous soils

Jawad Ali Shah and Guixin Chu

Department of Environmental Science and Engineering, College of Life Science, Shaoxing University, Shaoxing City, Zhejiang, P.R. China

Corresponding author
Guixin Chu, chuguixin@usx.edu.cn

## ABSTRACT

Short-chain polyphosphate fertilizers have been increasingly applied in agriculture, but little is known about the chemical behaviors of polyphosphate in soils. Herein, a cylinder experiment was carried out to investigate the influences of different P types (i.e., mono-ammonium phosphate (MAP), phosphoric acid (PA) and ammonium polyphosphate (poly-P)) and their application methods (single vs split) on the mobility and availability of P in soil through a column millimeter-scale slice cutting method; meanwhile a soil microcosm experiment (560-day) was conducted to investigate the effects of different P types on phosphorus dynamic transformation. Polyphosphate addition significantly increased P mobility. The average distance of P downward movement (81.5 mm) in soil profile in the poly-P application treatment increased by 33.6% and 81.1%, respectively, compared to the MAP and PA treatments. Different P application methods also markedly influenced phosphorus mobility. For instance, the average distance of P vertical movement in the split P application treatment was 21.2% higher than in the single application treatment, indicating that split P addition significantly increased P downward movement. Moreover, polyphosphate application decreased soil P fixation by blocking the transformation of the applied-P from labile to recalcitrant forms (HCl-P and residual-P). Overall, our findings provide meaningful information to current phosphorus fertilization practice in increasing soil P mobility and bioavailability. We suggest that polyphosphate could be regarded as an alternative P source used in agriculture, and split polyphosphate application is recommended as an effective P fertilization strategy.

## INTRODUCTION

Phosphorus (P) is an essential and yield-limiting macro-element for higher plants. However, about two-thirds of the total cultivated soil belongs to P-deficient soil (e.g., plant available $P$ <10 mg P kg$^{-1}$) in China (*Cao et al., 2012*; *Zhang et al., 2019*), and approximately 5.7 billion ha of area was deficient of soil available P on global scale

(*Hinsinger, 2001*). Chemically synthesized orthophosphate fertilizers (ortho-P) have been excessively applied in agriculture in the past several decades (*Chowdhury et al., 2017*). *MacDonald, Bennett & Taranu (2012)* addressed that P input was 23.8 Tg P·y$^{-1}$ globally in 2000, resulted in an annual P surplus of 11.5 Tg of P·y$^{-1}$. However, only a small fraction of the applied P was absorbed by plants (*Mueller et al., 2012*), and plant-accessible P only accounted for 6 % (range 1.5% to 11%) of total P in soils (*Stutter et al., 2012*). On the other hand, P is a non-renewable resource, the exploitable P ore will be depleted within 60–100 yr, because ca 31 million tons of P was exploited annually (*Cordell, Drangert & White, 2009*). Therefore, set up an effective P management strategy is imperative to recycle P more efficiently and avoid running out of the finite P resource.

The applied P in soils undergoes a sequence of intricate transformation processes. In neutral or calcareous soil, the freshly added P easily transformed from dibasic calcium phosphate (DCP) to brushite dicalcium phosphate dihydrate, (DCPD), then to octacalcium phosphate (OCP), finally to hydroxyapatite (HAP) (*Weihrauch & Opp, 2018*; *Ge et al., 2019*). When P is applied through basal application method, a relative high P concentration appears at P application site (*Chien et al., 2010*; *Li et al., 2019*). In this way, the added-P progressively converts to octa-calcium phosphate (Ca$_8$H$_2$(PO$_4$)$_6$·5H$_2$O, OCP) within 3 to 5 months, following by constitutes β-tri-calcium phosphate (β-TCP) after 8 to 10 months, finally turns into hydroxyapatite Ca$_{10}$(PO$_4$)$_6$(OH)$_2$. In contrast, more added P remains in available form when P fertilizer is split repeated applied at low application dose in crop growing season (*Khatiwada et al., 2014*; *González-Jimenez et al., 2019*). Consequently, a high crop yield and phosphorus use efficiency (PUE) can be achieved (*Saarela, Salo & Vuorinen, 2006*). Unfortunately, at present, basal application of P fertilizer remains used as main P fertilization technology in most agriculture areas.

Different P sources (ortho-P vs. poly-P) significantly influenced phosphorus availability and transformation (*McBeath et al., 2005*; *Wang & Chu, 2015*; *Gao et al., 2019*). Polyphosphates (poly-P) are superior over ortho-P fertilizers (*Hamilton et al., 2018*). This may largely be ascribed to its slow release characteristic (*Kulakovskaya, Vagabov & Kulaev, 2012*; *Hamilton, Hilger & Peak, 2016*). Following this, polyphosphate application significantly increased soil available P (*Gao et al., 2019*) and improved crop yield (*Holloway et al., 2004*). However, other studies showed that polyphosphate (ammonium polyphosphate, APP) had no obvious advantage or even less than ortho-P fertilizer (granular monoammonium phosphate, MAP) in increasing soil P availability and crop yield (*Ottman, Thompson & Doerge, 2006*). These controversial or paradoxical results implied that the physiochemical behaviors (i.e. mobility, availability, and transformation) of polyphosphate in soils need to be further explored.

In this study, a soil cylinder experiment and a soil microcosm experiment were carried out in contrasting calcareous soils. We hypothesize that split repeated polyphosphate application outcompetes conventional orthophosphate in increasing soil P availability and mobility. Therefore, the objectives of our study were to (i) compare the effects of different P types and their application methods on soil P mobility; (ii) clarify the response of soil P availability and transformation to different P addition, (iii) finally put forward an

**Table 1 Selected physical and chemical properties of the tested soils.**

| Soil properties | Loam | Clay |
|---|---|---|
| Texture[a] (<0.01 mm %) | 37.0 ± 0.02 | 71.4 ± 0.02 |
| pH[b] | 7.83 ± 0.03 | 8.15 ± 0.05 |
| EC[c] | 0.53 ± 0.01 | 1.22 ± 0.02 |
| OM[d] (g kg$^{-1}$) | 21.0 ± 0.23 | 15.6 ± 0.03 |
| Total-N[e] (g kg$^{-1}$) | 0.85 ± 0.05 | 0.71 ± 0.02 |
| Olsen-P[f] (mg kg$^{-1}$) | 24.2 ± 3.52 | 11.9 ± 2.18 |
| Water-P[g] (mg kg$^{-1}$) | 3.54 ± 0.39 | 3.58 ± 0.01 |
| Total-P[h] (g kg$^{-1}$) | 1.22 ± 0.12 | 0.95 ± 0.01 |
| Olsen-K[i] (mg kg$^{-1}$) | 518 ± 2.07 | 291 ± 5.53 |
| CaCO$_3^j$ (%) | 16.1 ± 0.00 | 15.0 ± 0.00 |

Notes:

Data were presented as the mean ± standard deviation (SD), $n = 3$ at a significance level of $p < 0.05$.

[a] Texture was determined by the Bouyoucos hydrometer method and Katschinski classification System (*Bouyoucos, 1962*).

[b] pH was determined at soil to milli-Q water ratio of 1:2.5 w/v using pH meter.

[c] Soil EC was measured using a 1:2.5 ratio of soil to Milli-Q water.

[d] Organic C was measured by the wet-oxidation technique (*Shaw, 1959*).

[e] Soil total N was measured by the semimicro-Kjeldahl method; (*Bao, 2000*).

[f] Olsen P was measured by the Olsen method (*Olsen et al., 1954*).

[g] Water-P was measured using a 1:25 ratio of soil to milli-Q water.

[h] Total P was measured by the perchloric acid digestion method (*Olsen et al., 1954*).

[i] Olsen K was measured by the flame photometry method (*Bao, 2000*).

[j] CaCO$_3$ was measured by neutral titration method (*Bao, 2000*).

appropriate P fertilization strategy to increase soil P availability and improve PUE in calcareous soil.

# MATERIALS AND METHODS

## Soils sampling and soil description

Experiments were conducted using two calcareous soils (loam and clay soils). According to FAO/WRB soil taxonomy, both were classified as Calcisol Fluvisols. Loam soil was taken from experimental station of Shihezi University (44°18′ N, 86°02′ E) and clay soil was collected from 147 State Farm (44°37′ N, 86°10′ E) in Shihezi region. Soil samples were air dried, stones and small visible plant residues were manually removed. All samples were ground to pass through a 2 mm sieve prior to measurement of soil properties. The selected soil physical and chemical properties are presented in (Table 1).

## Different P sources description

Three types of P sources including (i) mono-ammonium phosphate (powder MAP, $NH_4H_2PO_4$) with purity of 99% and density of 1.803 g cm$^{-3}$ and 26.9% of P (Shengao Chemical Reagent Co., Ltd., Shanghai, China); (ii) phosphoric acid (fluid PA, $H_3PO_4$) with purity of 85% and density of 1.685 g cm$^{-3}$ and 27.1% of P (Fuyu Special Chemicals Co., Ltd., Dongying, China), and (iii) ammonium polyphosphate (fluid APP, $H_6P_4O_{13}$) with purity of 85% and density of 2.1 g cm$^{-3}$ and 31.3% of P (Aladdin Industrial Corporation, Mooresville, NC, USA) were employed in this study.

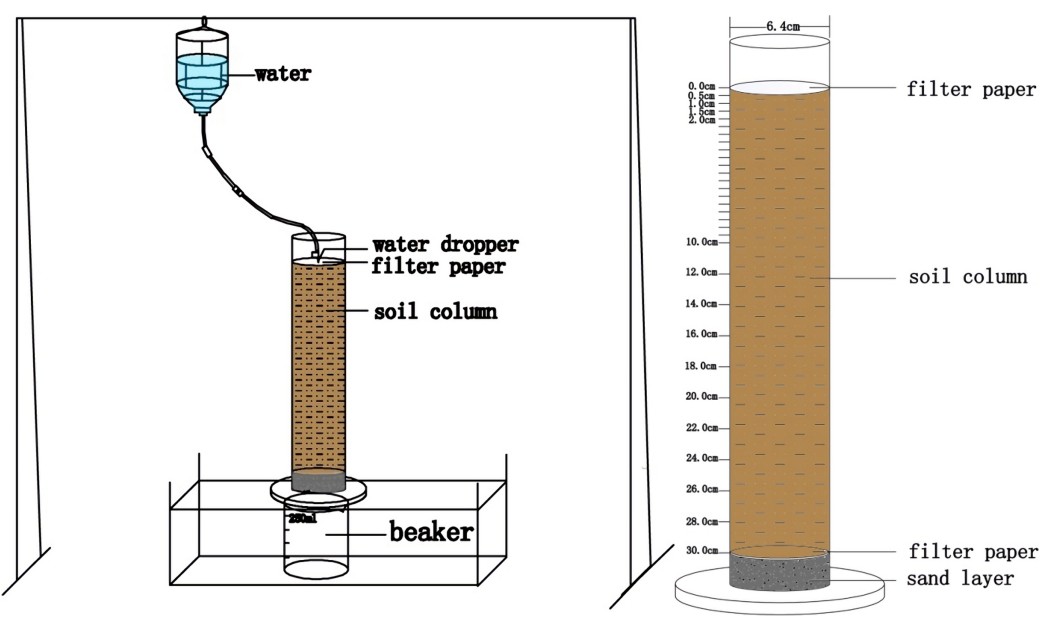

**Figure 1 Schematic diagram of soil cylinder experiment.**

## Experiment I soil cylinder experiment

Soil cylinder experiment was carried out in a transparency Plexi glass cylinders (350 mm height, 64 mm internal diameter). Five holes (2 mm diameter) were drilled at bottom of the cylinder to maintain aerobic condition. A 10-mm-thick layer of fine sand was placed at the bottom of the cylinder and placed two pieces of filter paper on the surface of the sand layer. A total of 1,302 g loam soil and 1,331 g clay soil were placed into each cylinder. Another two pieces of filter paper was placed on the top of the cylinder. Soil bulk densities of 1.35 g cm$^{-3}$ and 1.38 g cm$^{-3}$ were designed for loam and clay soil, respectively. Three P types (i.e., powder MAP, fluid PA, and fluid APP) were applied to the each cylinder following the methods of: (i) Single application, the total amount of P fertilizer was uniformly mixed into the upper 20 mm of soil layer and watered four times over the 4-week-long incubation period; (ii) split repeated application, the fertilizer was initially dissolved with 50 mL milli-Q water in a 250 mL beaker individually except MAP which was applied directly and then fertilizer solution was injected into soil at the center site of the cylinder using a syringe, P fertilizer was four times evenly added at equivalent ratio during entire 28-days period. The irrigation water was poured into a Markov bottle, and hanged the bottle 1.5 m above each cylinder (Fig. 1). The flow rate of the emitter was controlled at 6 drops per second by connecting each emitter to individual Markov bottle. The total water column irrigated were 401.6 mL and 515.8 mL cylinder$^{-1}$ for loam and clay soils, respectively. P application rates were 58.5 mg P cylinder$^{-1}$ for loam and 59.8 mg P cylinder$^{-1}$ for clay soils, respectively. The P fertilizer application rate was two to three times higher than in normal farmland (*Du et al., 2013*; *Wang & Chu, 2015*). Each treatment replicated three times giving a total of 42 cylinders (containing 6 CK treatments with no fertilizer either in loam or clay soil, 18 cylinders

for single application of MAP, PA and APP either in loam or clay soil and another 18 cylinders for split application of MAP, PA and APP either in loam or clay soil). All cylinders were covered with a piece of parafilm, and incubated at room temperature during incubation period.

After 28-day incubation, all cylinders were placed into a −80 °C freezer for 12 h to keep soil column hard enough. The frozen cylinders were cut into the given thick soil slices by placing the cylinder to a high-speed spindle of the lathe with a sharp blade at millimeter-scale (Type: CW6163C, made in Dalian, China) (Supplementary Figures). Each cylinder was vertically cut into 20 slices with a 5-mm thickness for each slice, then another 10 slices with a 20-mm-thickness were cut. Each slice was immediately placed in a plastic bag and transferred to fridge.

All collected soil slices were used to determine soil moisture, soil WE-P (water extracted phosphorus) and Olsen-P. Briefly, to determine WE-P, an aliquot 2 g soil was taken in a 100 mL centrifuge tube with 50 mL triple de-ionized (TDI) water. Shaking the soil sample for 1 h at 25 °C, following centrifugation for 15 min at 900 rpm, decant supernatant was separated and WE-P was measured using malachite green colorimetric method at an absorbance of 610 nm (Masson et al., 2001). Olsen-P was determined in 0.5 M NaHCO₃ extracts (pH 8.5), using the molybdenum blue method (Olsen et al., 1954).

An exponential decay equation model was employed to describe the P downward movement in soil cylinder.

$$Y = (Y_0 - \text{Plateau}) \times \exp(-K \times X) + \text{Plateau} \tag{1}$$

where X represents the depth of the soil column expressed as mm. Y indicates the content of phosphorus in soil column in mg/kg. $Y_0$ is the Y value when X (depth) is zero. It is expressed in the same units as Y; Plateau is the Y value at infinite depths, expressed in the same units as Y; K is the rate constant, expressed in reciprocal of the X axis depth units ($mm^{-1}$); Half-depth is in the depth units of the X axis. It is computed as ln(2)/K, which means the movement depth when the concentration of P drop down to the half.

**Experiment II soil incubation experiment**

To explore the influences of different P types on phosphorus transformation, a 560-day microcosm incubation experiment was set up. Herein only loam soil was used as described in the previous section. In brief, a total of 253 g soil was thoroughly mixed with different P fertilizers (MAP, PA and APP) at the addition rate of 40.3 mg P $pot^{-1}$. The mixed soil was placed in a plastic pot (5.7 cm in height, 6.6 cm in diameter) with soil bulk density of 1.30 g $cm^{-3}$, all pots were incubated under room condition for 560 days. Tap water was added periodically per week to keep water holding capacity of 60% (WHC). Each treatment was replicated four times giving a total of 16 pots (including 4 CK treatments having no fertilizer).

After 65-, 140-, 230-, 320-, 560-day incubation, soil samples were collected from each pot. All soil samples were air dried and grounded for determination of different soil inorganic P fractions according to Guppy et al. (2000) and modified by Aulakh et al. (2003). Briefly, soil P sequential extraction followed: (1) resin-P: an aliquot 0.500 g soil sample

was mixed with 30 mL deionized water in a 500 mL centrifuge tube, two Cl-saturated anion exchange resin strips were added to remove resin-P, the sample was shaken for 16 h. After shaking the two Cl-saturated anion exchange resin strips were taken in a 100 mL centrifuge tube and 30 mL 0.7 moL/L NaCl was added subsequently, shaking the sample for 1 h. (2) $NaHCO_3$-P: the soil residue in the previous step was treated with 30 mL 0.5 moL/L $NaHCO_3$ to remove $NaHCO_3$-P, shaking the sample for stay overnight, and centrifugation for 30 min. (3) NaOH-P: the soil residue was treated with 30 mL 0.1 M NaOH and 1 mL of 4 M NaCl to remove NaOH-P, shaking the sample for 16 h, and subsequently centrifugation for 30 min. (4) HCl-P: the soil residue was treated with 30 mL 1moL/L HCl, shaking stay over. (5) Residue-P: aliquot 0.5 g anhydrous $MgSO_4$ and 5 mL of $H_2SO_4$: $HClO_4$ (20:1) acid solution to remove residue-P. Inorganic P concentration in each of the extracts was determined using malachite green colorimetric method at an absorbance of 610 nm (*Masson et al., 2001*). The total P content of soil samples was determined using molybdenum blue spectrophotometry method after digestion with $HClO_4$–$H_2SO_4$ (*Masson et al., 2001*).

The proportion of applied P fertilizer transformed to different P fractions in soils was figured out referring to *Aulakh et al. (2003)*. Individual parameters were calculated as

Amount of fertilizer P recovered as Pi in soil (kg P $ha^{-1}$)
$$= [\text{amount of Pi in P} - \text{fertilized treatment (kg P } ha^{-1})]$$
$$- [\text{amount of Pi in no} - \text{P control (kg P } ha^{-1})] \tag{2}$$

Total P recovered in soil (kg P $ha^{-1}$)
$$= [\text{sum of P recovered in different fractions in the soils (kg P } ha^{-1})$$
$$\times (2.325 \times 106 \text{ kg } ha^{-1}) \tag{3}$$

where $2.325 \times 10^6$ kg P $ha^{-1}$ is soil mass of 0–150 mm soil layer computed using field bulk density of 1.55 g $cm^{-3}$.

Total fertilizer P recovered in soil (kg P $ha^{-1}$)
$$= [\text{total P recovered in P} - \text{fertilized treatment (kg P } ha^{-1})]$$
$$-[\text{total P recovered in no} - \text{P control (kg P } ha^{-1})] \qquad ] \tag{4}$$

Fertilizer P present in any specific soil fraction (%)
$$= \frac{[(\text{Pi in P} - \text{fertilized treatment }) - (\text{Pi in no} - \text{P control })](kgPha^{-1})}{\text{Total fertilizer P recovered in soil (kg P } ha^{-1})} \tag{5}$$

## Statistical analyses

Data were analyzed using the SPSS 11.5 statistical program (SPSS Inc., Chicago, IL, USA) with two-way ANOVA at a significance level of $p < 0.05$. A Duncan multiple range test was carried out to test the significant differences between different treatments. Microsoft Excel 2003 and Graphpad Prism 5.0 software (GraphPad Software, Inc., San Diego,

CA, USA) were used for data processing and images making. All results in figures and tables were presented as mean of three or four replicates with a standard deviation (SD), a statistical significance level of $p < 0.05$ was used for all analyses.

## RESULTS

### Different P application methods on P mobility and availability

Soil Olsen-P decreased with soil depth (0–100 mm) increasing across all P sources (Fig. 2). Compared to the single application treatment, the significant higher values of Olsen-P were always appeared in the repeated PA and APP application at 0–40 mm depth ($p < 0.05$). The vertical movement of Olsen-P reached the same value with CK (baseline) were 80 mm and 95 mm, respectively, in the PA and APP repeated application treatment, while those were merely 42 mm and 50 mm in the single application treatment in loam soil. Similar tendency was also observed in clay soil, indicating that repeated P application significantly promoted P vertical migration.

An exponential decay model was employed to describe Olsen-P movement in soil cylinder (Table 2). Rate constant (K) and half-depth were key parameters to reflect the effects of different application methods on P movement. The steeper the response curve (higher K value) and lower half-depth, the shorter distance the added-P moved, and vice versa. In clay soil, K value (Rate constant) in the repeated P application was lower than that in the single application treatment across all treatments. The values of half-depth increased by 39.7%, 49.8%, 158.6% in the repeated application treatment, respectively, relative to the single MAP, PA and APP application treatments.

### Different P types on P mobility and availability

Compared to the treatments of MAP and PA application, soil P mobility was significantly increased by APP application (Fig. 3). The distance of WE-P downward movement in the single P application treatment in loam soil followed order of APP (80 mm) > MAP (60 mm) > PA (35 mm). Similarly, the distance of WE-P downward movement in clay soil followed the order of APP (83 mm) > MAP (62 mm) > PA (55 mm). Moreover, the exponential decay equation model showed that K value in loam soil followed the order of APP ($0.054\ mm^{-1}$) < MAP ($0.065\ mm^{-1}$) < PA ($0.115\ mm^{-1}$), and the half-depth followed the order of APP (12.70 mm) > MAP (10.66 mm) > PA (6.02 mm) when P fertilizer applied as single application method (Table 3). Likewise, K value in the repeated application treatment (loam soil) followed order of APP ($0.039\ mm^{-1}$) < MAP ($0.044\ mm^{-1}$) < PA ($0.062\ mm^{-1}$), the half-depth followed the sequence of APP (17.92 mm) > MAP (15.88 mm) > PA (11.24 mm). The half-depth in the APP treatment were 4.84% and 31.5%, respectively, greater than the MAP and PA treated clay soils. These results indicate that APP fertilization significantly improved P mobility and availability.

### Inorganic P transformation

During whole incubation time, resin-P and $NaHCO_3$-P (labile P) notably decreased, while NaOH-P (moderately labile P) and HCl-P (recalcitrant soil P) showed an increasing tendency (Fig. 4). Furthermore, soil resin-P, $NaHCO_3$-P and NaOH-P in the APP added

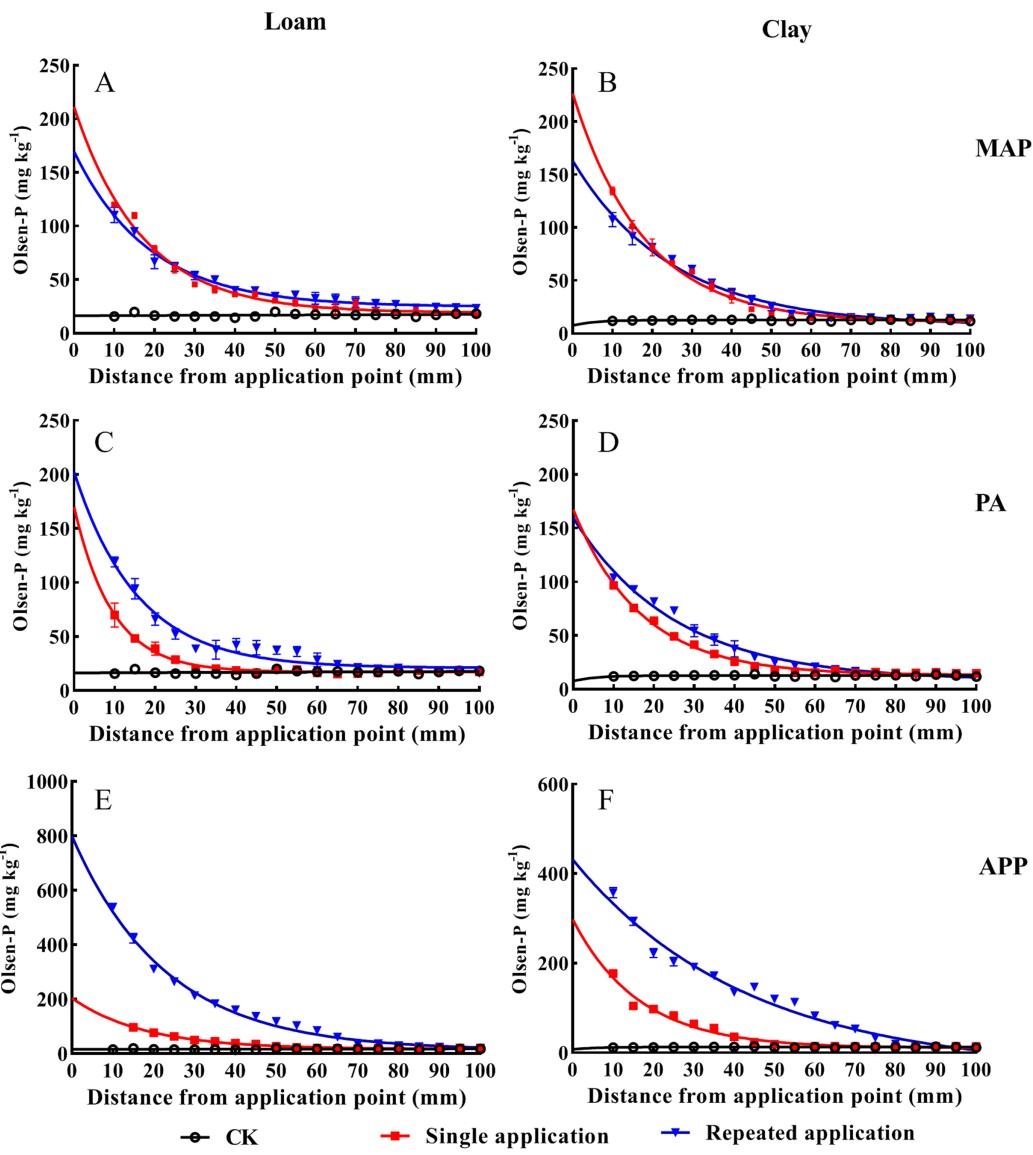

**Figure 2** **The effect of different P application methods on phosphorus mobility and availability.** Data are presented as mean ± standard deviation (SD) of three replicates at a significance level of $p < 0.05$. The three lines represent MAP, PA and APP, respectively. The left column represents loam soil; the right column represents clay soil. Vertical bars represent standard deviation of the mean ($n = 3$). The black line with circle represents CK, the red line with square frame represents single P fertilization and the blue line with triangle represents fertilizers repeated application treatment.

treatment were significantly higher than in the PA and MAP treatments. For instance, at the end of incubation (560 d), resin-P (5.33 mg kg$^{-1}$) and NaHCO$_3$-P (73.7 mg kg$^{-1}$) in the APP addition treatment increased by 51.6% and 191% respectively, relative to the MAP and PA application treatments (on average). In addition, When P fertilizer was added as polyphosphate, NaOH-P was increased by 24.6% and 39.8%, respectively, relative to the MAP and PA treatments. The influence of the different P types on non-labile P

**Table 2 Exponential equations simulating the influences of different P fertilization methods on soil P downward movement (Olsen-P, mg kg$^{-1}$).**

| P form | Application method | Loam soil | | | | Clay soil | | | |
|---|---|---|---|---|---|---|---|---|---|
| | | Equation | K (mm$^{-1}$) | Half-depth (mm) | R$^2$ | Equation | K (mm$^{-1}$) | Half-depth (mm) | R$^2$ |
| MAP | Single | $y = 192.03e^{-0.059x} + 19.07$ | 0.059 | 11.8 | 0.97 | $y = 216.05e^{-0.056x} + 10.45$ | 0.056 | 12.49 | 0.98 |
| | Repeated | $y = 144.70e^{-0.053x} + 24.70$ | 0.053 | 13.1 | 0.94 | $y = 155.72e^{-0.040x} + 7.38$ | 0.040 | 17.45 | 0.96 |
| PA | Single | $y = 152.68e^{-0.105x} + 17.12$ | 0.105 | 6.57 | 0.87 | $y = 153.98e^{-0.059x} + 13.12$ | 0.059 | 11.72 | 0.98 |
| | Repeated | $y = 181.36e^{-0.063x} + 20.94$ | 0.063 | 10.9 | 0.90 | $y = 151.40e^{-0.040x} + 8.10$ | 0.040 | 17.56 | 0.97 |
| APP | Single | $y = 187.05e^{-0.056x} + 17.25$ | 0.056 | 12.3 | 0.97 | $y = 286.92e^{-0.060x} + 10.58$ | 0.060 | 11.51 | 0.81 |
| | Repeated | $y = 785.95e^{-0.044x} + 12.65$ | 0.044 | 15.9 | 0.99 | $y = 471.57e^{-0.023x} - 40.17$ | 0.023 | 29.77 | 0.94 |

Note:
    Data were presented as the mean value of three replicates and standard deviation (SD) at a significance level of $p < 0.05$.

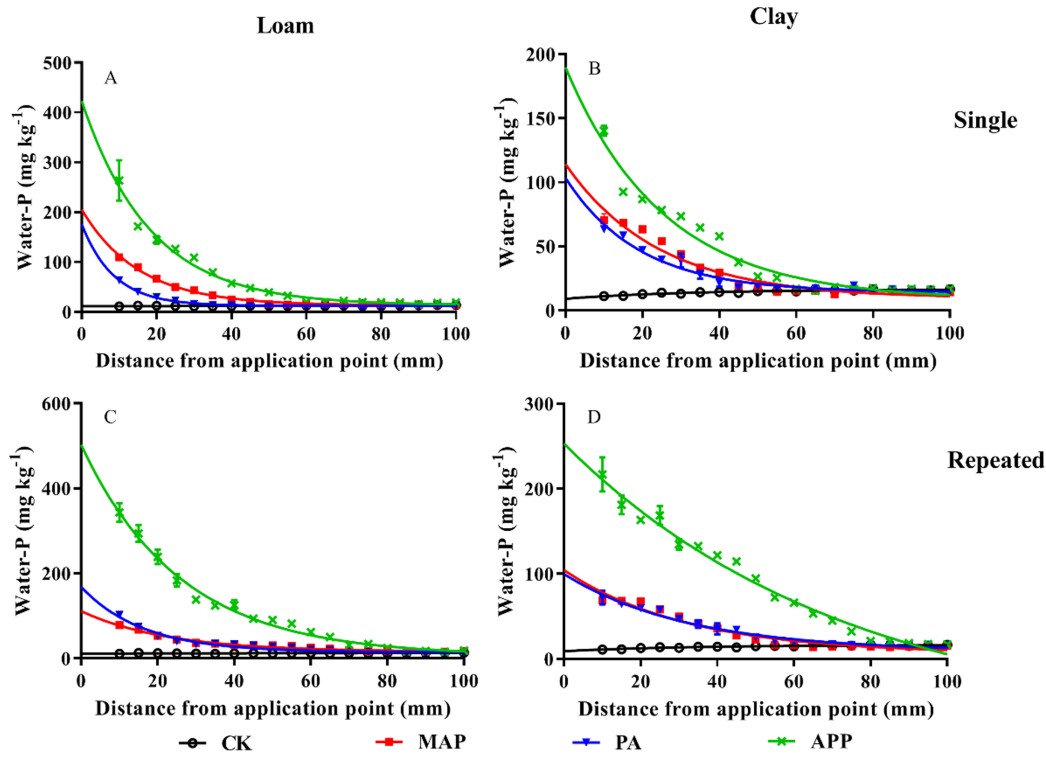

**Figure 3 The influences of different P sources on soil phosphorus mobility and availability.** Data are presented as mean ± standard deviation (SD) of three replicates at a significance level of $p < 0.05$. The upper and lower rows represent single application and repeated application method, respectively. The left line represents loam soil; the right line represents clay soil. Vertical bars represent standard deviation of the mean ($n = 3$). The black line with circle represents CK, the red line with square frame represents MAP, the blue line with triangle represents PA, and the green line with cross represents APP treatment.

distribution proportion followed order of MAP (87.8%) > PA (73.7%) > APP (54.5%) (Table 4), indicating that polyphosphate addition significantly retarded the transformation of the added-P from labile to non-labile P forms, thus reduced P fixation.

**Table 3 Exponential equations simulating the influences different P sources on phosphorus downward movement (water-P, mg kg$^{-1}$).**

| Application method | P form | Loam soil | | | | Clay soil | | | |
|---|---|---|---|---|---|---|---|---|---|
| | | Equation | K (mm$^{-1}$) | Half-depth (mm) | R$^2$ | Equation | K (mm$^{-1}$) | Half-depth (mm) | R$^2$ |
| Single | MAP | $y = 192.21e^{-0.065x} + 12.79$ | 0.065 | 10.7 | 0.97 | $y = 104.94e^{-0.041x} + 9.56$ | 0.041 | 16.72 | 0.94 |
| | PA | $y = 163.38e^{-0.115x} + 12.02$ | 0.115 | 6.02 | 0.92 | $y = 89.25e^{-0.052x} + 14.25$ | 0.052 | 13.33 | 0.93 |
| | APP | $y = 410.54e^{-0.054\ x} + 12.96$ | 0.054 | 12.7 | 0.95 | $y = 182.075e^{-0.040x} + 8.53$ | 0.040 | 17.53 | 0.96 |
| Repeated | MAP | $y = 95.8e^{-0.044x} + 15.30$ | 0.044 | 15.9 | 0.92 | $y = 98.48e^{-0.031x} + 6.02$ | 0.031 | 22.43 | 0.93 |
| | PA | $y = 153.71e^{-0.062x} + 14.29$ | 0.062 | 11.2 | 0.93 | $y = 90.86e^{-0.031x} + 8.94$ | 0.031 | 22.47 | 0.91 |
| | APP | $y = 496.43e^{-0.039x} + 5.87$ | 0.039 | 17.9 | 0.97 | $y = 334.01e^{-0.014x} - 80.51$ | 0.014 | 50.93 | 0.96 |

Note:
Data were presented as the mean value of three replicates and standard deviation (SD) at a significance level of $p < 0.05$.

**Table 4 The influences of MAP, PA and APP applications on different inorganic P fractionations in soil (after 560-day incubation).**

| Treatment | Resin-P (%) | NaHCO$_3$-P (%) | NaOH-P (%) | HCl-P (%) | Residue-P (%) |
|---|---|---|---|---|---|
| MAP | 0.23 ± 0.06 | 6.13 ± 0.2 | 5.82 ± 0.8 | 21.7 ± 0.6 | 66.1 ± 1.0 |
| PA | 1.07 ± 0.03 | 20.0 ± 0.1 | 5.19 ± 0.1 | 58.2 ± 0.9 | 15.5 ± 0.8 |
| APP | 1.64 ± 0.15 | 36.7 ± 0.4 | 7.25 ± 0.9 | 45.4 ± 0.5 | 9.10 ± 0.1 |

Notes:
MAP, mono-ammonium phosphate; PA, phosphoric acid; APP, ammonium polyphosphate.
Data were presented as the mean value of four replicates and standard deviation (SD) at a significance level of $p < 0.05$.

## DISCUSSION

Generally, short-chain soluble polyphosphate fertilizers are made up from different inorganic P components at a certain proportion (i.e. ortho-P, pyro-P, triple-P, tetra-P). Therefore, poly-P can't be directly absorbed by plants unless it be hydrolyzed into orthophosphate ($H_2PO_4^-$ or $HPO_4^{2-}$) (*Kulakovskaya, Vagabov & Kulaev, 2012*). In this study, phosphorus downward movement in soil column was significantly influenced by different types of P fertilizers (Table 2; Fig. 3). Application of slow release fertilizers (poly-P) significantly increased P movement. We found that soil P vertical movement well-fitted the nonlinear exponential regression ($0.81 < R^2 < 0.99$), which provides a reliable evidence to our hypothesis that split repeated polyphosphate application was superior over the single application method in increasing P mobility. Following this, when P fertilizer is applied in single basal method, the freshly added-P is easily fixed due to high P concentration in fertilization placement (*Yang et al., 2012*). A in situ imaging of liquid-cell atomic force microscopy (AFM) showed that the formation of calcium phosphates (Ca-P) nanoclusters was markedly increased as P presented at high concentration [50 mM $(NH_4)_2HPO_4$] (*Wang & Putnis, 2020*). Besides, the direction of P diffusion from P application site to outside is opposite to the direction of water movement caused by low soil water potential (*Hedley & McLaughlin, 2005*). Therefore, the diffusion of $H_2PO_4^-/HPO_4^{2-}$ would inevitably be blocked by water movement (*Hinsinger, 2001*). In contrast, when P fertilizer is applied through split repeated method, the

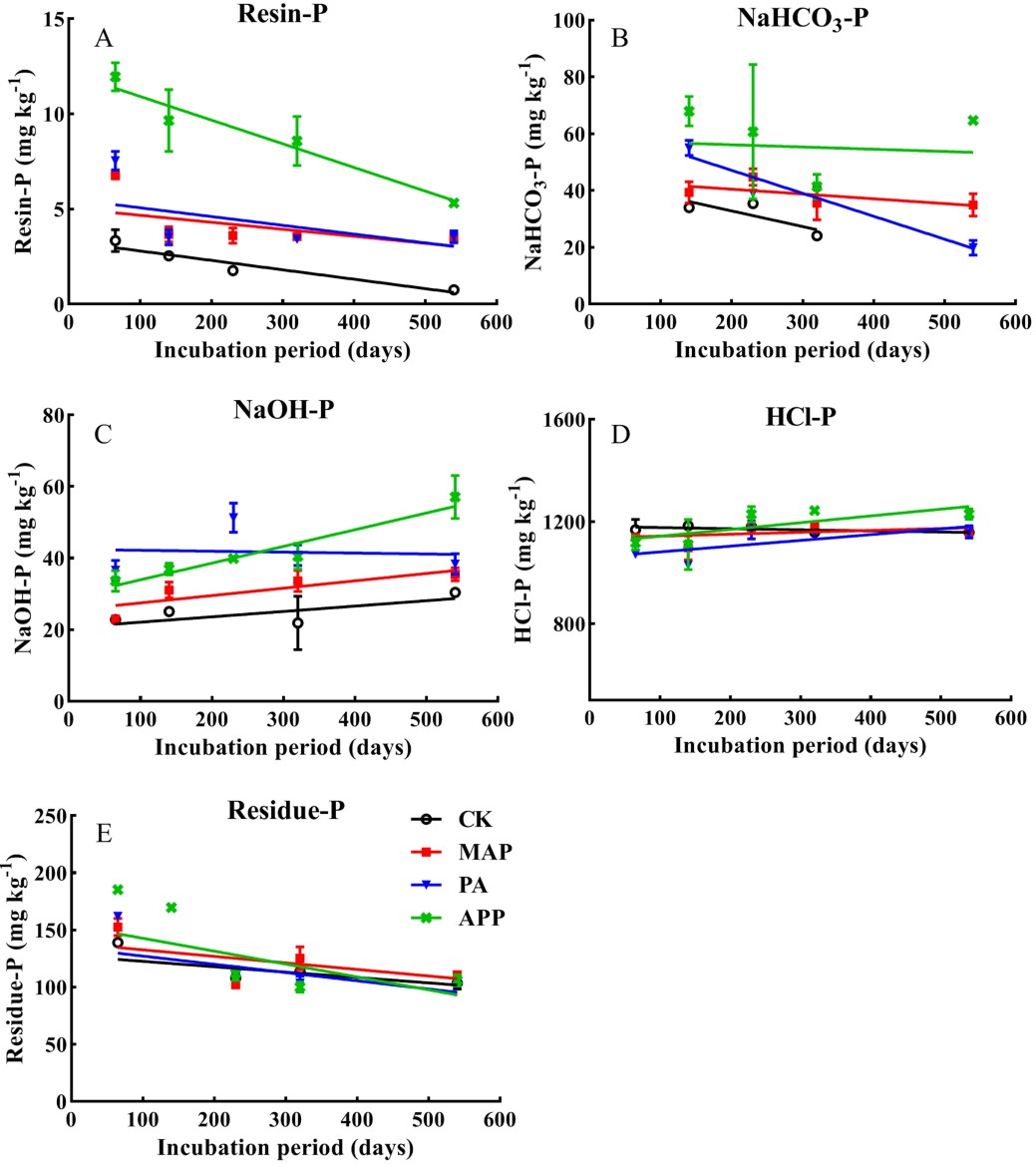

**Figure 4 The effect of different P sources and their application methods on the changes of soil resin-P, NaHCO₃-P, NaOH-P, HCl-P and residue-P.** Data are presented as mean ± standard deviation (SD) of four replicates at a significance level of $p < 0.05$ ($n = 4$). The black line with circle represents CK, the red line with square frame represents MAP, the blue line with triangle represents PA, and the green line with cross represents APP treatment.

concentration of P in P application site is relative low, resulted in the distribution of P in soil was scattering and dispersive (*Hettiarachchi et al., 2006*; *McLaughlin et al., 2011*). Our results were consistent with the findings by *Lombi et al. (2004)* who noted that fluid sources of P enhanced phosphorus mobility and liability compared with granular P fertilizers. The reasons may likely attributed to: (i) poly-P fertilizers belong to slow-release fertilizer, they move in condensed form in soil before it be hydrolyzed (*McBeath et al., 2007*), and it hardly interacted with soil mineral particles of $Fe_2O_3/Al_2O_3$ (*Hamilton, Hilger & Peak, 2016*; *Hamilton et al., 2018*); (ii) poly-P may mobilize soil native P via chelating

**Table 5  pHs in MAP, PA and APP fertilizer and mixed with soils.**

| Treatment | pH (fertilizer) | pH (fertilizer mixed in loam soil) | pH (fertilizer mixed in clay soil) |
|---|---|---|---|
| MAP | 5.15 ± 0.08 | 7.81 ± 0.07 | 8.37 ± 0.03 |
| PA | 2.01 ± 0.05 | 7.99 ± 0.09 | 8.38 ± 0.09 |
| APP | 1.36 ± 0.02 | 7.55 ± 0.06 | 8.06 ± 0.05 |

Notes:
MAP: mono-Ammonium phosphate; PA: phosphoric acid; APP: ammonium polyphosphate.

reaction. In this way, McBeath et al. (2007) showed that poly-P activated recalcitrant oxidation forms of soil P ($Fe_2O_3$ and $MnO_2$). Similarly, Wang et al. (2019) found that poly-P fertilizer treatment significantly increased P availability in calcareous soil. Besides, poly-P fertilizer application decrease soil rhizosphere pH value at 0.1–0.5 unit (Hamilton et al., 2018; Rahmatullah, Wissemeier & Steffens, 2006; McBeath et al., 2007; Du et al., 2013). Likewise, our data showed that the poly-P fertilizer application treatment decreased solution pH by ~0.34 unit than MAP and PA fertilizer (Table 5), which might be another possible reason for polyphosphate mobilizing native soil P. Although the effects of poly-P applied to agricultural soils on increasing soil P availability and reducing its fixation is promising, nevertheless further study is needed by using edge-cutting technologies to fully elucidate the mechanisms of mobilizing effect by poly-P application and their interaction with soil particles in a wide-range of soil conditions.

## CONCLUSION

The mobility, effectiveness, and availability of fertilizer P were significantly increased with ammonium-polyphosphate application compared with MAP and PA application. Compared to P fertilizer basal application, split repeated P application markedly promoted soil P movement. Moreover, the added-P transformation from labile (resin-P and $NaHCO_3$-P) to non-labile forms (HCl-P and residue-P) in the poly-P treated soil was significantly retarded, in contrast to MAP and PA application treatments. Therefore, considering the scarcity of P resource and low P fertilizer use efficiency, polyphosphate fertilizers coupled with split repeated application method is recommended as an effective P management strategy in increasing soil available P and decreasing P fixation in calcareous soil.

## ACKNOWLEDGEMENTS

Thanks are given to Dr. & Prof. Wakelin A Steve from New Zealand for his helpful comments and revision.

### Funding

This work was jointly supported by National Natural Science Foundation of China (No. 41161047), and the scientific development and technology innovation project of Xinjiang Production and Construction Group (XPCG), China (No. 2017BA041). The funders had

no role in study design, data collection and analysis, decision to publish, or preparation of the manuscript.

## Grant Disclosures
The following grant information was disclosed by the authors:
National Natural Science Foundation of China: 41161047.
Xinjiang Production and Construction Group (XPCG), China: 2017BA041.

## Competing Interests
The authors declare that they have no competing interests.

## Author Contributions
- Jawad Ali Shah performed the experiments, analyzed the data, prepared figures and/or tables, and approved the final draft.
- Guixin Chu conceived and designed the experiments, performed the experiments, analyzed the data, prepared figures and/or tables, authored or reviewed drafts of the paper, and approved the final draft.

## Data Availability
The raw data are available in the Supplemental Files.

## Supplemental Information
Supplemental information for this article can be found online at http://dx.doi.org/10.7717/peerj.11493#supplemental-information.

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
