# Peer review of "Short-chain soluble polyphosphate fertilizers increased soil P availability and mobility by reducing P fixation in two contrasting calcareous soils"

_PeerJ, doi:10.7717/peerj.11493_

## Round 0.1 · original submission · Major Revisions

The paper needs major revision. Revise your paper according to the reviewers' comments and resubmit.

Reviewer 1 ·

Basic reporting

- Overall, the text is well written, yet a few point of attention are:
o Line 62: parenthesis before DCPD
o Line 75: “Some studies showed short-chain (n<20) soluble polyphosphates (poly-P) superior over ortho-P fertilizers in increasing crop productivity”, should be written as “polyphosphates (poly-P) are superior over ortho-P fertilizers” or “polyphosphates (poly-P) to be superior over ortho-P fertilizers”
o Line 76: “This may largely ascribe to its slow release characteristic” should be written as “This may largely be ascribed to its slow release characteristic”
o Line 89: a “iii)” should be added here fore the last point
o Line 212: “repeated P application significantly promoted P vertical migrant” should be written as e.g. ““repeated P application significantly promoted vertical P migrantion”
o Line 140: “phosphorous”; Line 157, Line 255: “phosphorus”. Try to be consistent with the use of the P abbreviation.
o Line 284: “their interact with soil particles in a wide-range of soil conditions.” Interact should be changed to interaction.
- A lot of relevant literature has been cited in the introduction, yet the authors should revise the statements that were made (see also comments on validity of the findings).
- Figures are clear and relevant.
- Line 116: The method used to apply the single fertilizer application is not explained clearly. “The amount of P fertilizer was uniformly mixed with the upper 20 mm of soil” but also the authors write that “the P solutions were sprayed on the upper soil surface of the soil cylinder”.
- Line 287: “In summary, our study showed that ammonium-polyphosphate application significantly increased soil phosphorus mobility and availability”. Sentence should be revised: the phosphorus mobility and availability increased compared to what?
- The raw data are presented very clearly.

Experimental design

- Line 133: “all cylinders were placed into a -80℃ freezer for 12 hours”. Why was opted for this freezing method? Freezing of the column is generally not required to sample the soil in the column along with the column depth. Can the freezing affect the P speciation, e.g. by concentrating the solution P concentration during the freezing step, or by slightly changing the pore structure? Has this method been used before in comparable work?
- Soil cylinder experiment: No pH measurement was performed on the obtained soil samples from the column, although acid-base effects are expected with the use of the reported P fertilizer compounds. The obtained soil pH can also strongly affect to the availability and mobility of the fertilizer P and the soil P.
- An important issue that was not addressed sufficiently in this work, either in the introduction or in the experiments, is the application form of the fertilizers. For example, the MAP fertilizer was dissolved prior to application, while most literature studies cited in the manuscript report on fertilizer trials with granular MAP, which is also the most commonly used form of this fertilizer. Why did the authors opt to dissolve the MAP prior to application?

Validity of the findings

- Figure 2: in the loamy soil, it seems like there is a data point missing for the single APP treatment at 10 mm? How does this affect the modeling, and the statement made on line 207, as for example the single application of MAP at 10 mm depth could have a quite comparable value for Olsen P?
- A main concern on the interpretation of the results of this study refers again to the application form of the fertilizer MAP, and more attention should be given to this in the discussion. Also, the authors refer to literature findings in the discussion, that clearly relate to a granular MAP application (e.g. Hedley & McLaughlin, 2005 and by Lombi et al., 2004), while no granular fertilizers were tested in this study. Therefore, I believe the interpretation of the results of this study should be done more carefully with respect to not only the P fertilizer source but also the application form (dissolved MAP in this study, granular MAP in many literature studies).
- In line with the previous comment, the authors should revise the literature statements that are made in the manuscript, because the corresponding citations do not always confirm the statements made in the manuscript. In that sense, the discussion definitely still needs to be improved.
- The study investigates both P availability and P mobility from different P fertilizer sources and application methods. The introduction focusses on the efficiency (PUE) of polyphosphate fertilizer relative to conventional orthophosphate fertilizers. The matter of P mobility (P leaching) is, however, not addressed. A high P availability is obviously a positive aspect for fertilizer efficiency; however, a high mobility of P in the soil is not always beneficial for the PUE. Naturally, from a chemical perspective, there is a link between P availability and P mobility, but also this is not discussed. Could the authors elaborate more on the importance of both P availability and P mobility of the different P fertilizer sources?

Additional comments

- Line 281: “Therefore, future study should combine using edge-cutting technologies such as atomic force microscopy (AMF), Raman spectrometer and Fourier transform infrared spectrometer (FTIR) and X-ray absorption near edge structure (XANES) to provide in situ direct evidence regarding the behaviors of polyphosphate fertilizer and their interact with soil particles in a wide-range of soil conditions.”. This statement is too broad. What information can still provide additional knowledge on the behavior or P and poly-P in calcareous soils? What samples are interesting to be analyzed with which techniques?

·

Basic reporting

English Editing required
Title did not express the body of article I suggest to revise it.
Article structure figures and tables need improvement
Lack self contained with relevant results to hypothesis

Experimental design

The author is talking about mobility of P in Soil But I did not see hydraulic properties of the soil in text. Please provide the Hydraulic properties of Soils after experimental Set Up. The soil column develop with disturbed soil which ultimately affect the Hydraulic properties of soils. So, the mobility of P with out soil Hydraulic properties is under question?
2ndly, the solubility of the different P sources is missing in text. Please provide the solubility rate.
Methods need improvement for replication of experiment.
There is need to correlate the identified knowledge gap with findings

Validity of the findings

The detail about the statistical tests they performed on the data before doing the ANOVA since it is a parametric test, normality and homogeneity of variance should be verified.

Additional comments

This study is of great importance. The Introduction section is well written with recent references. I appreciate the fact that the authors described the methodology used in detail in the study. It helps to understand the results section thoroughly. However, I have some concerns about the different parts of the manuscript. Moreover, the paper lacks many major details. I suggest a major revision to address a few issues. If the authors address carefully the comments, I’ll recommend publication of the manuscript in the journal. Please, do include more implications of the results of the study in the conclusion and abstract sections.

---

## Round 0.2 · accepted · Accept

The manuscript is revised and improved a lot. I checked the manuscript and found it better to be accepted for publication in PeerJ, thanks.

Reviewer 1 ·

Basic reporting

The new title for the paper suggested by another reviewer is, in my opinion, definitely not an improvement. With this new title, it is not clear what the reference situation is for the ‘increased’ availability and mobility of short-chain soluble polyphosphate fertilizers, while in the previous title this was well stated (“Short-chain soluble polyphosphate outperforms orthophosphate fertilizers…”). I believe it is important to somehow mention in the title that this study compares polymeric phosphate fertilizer with orthophosphate fertilizer.

Experimental design

No comment

Validity of the findings

No comment

Additional comments

The issues I addressed in the previous version of the manuscript were sufficiently addressed in the revised version, I thank the authors for the changes that were made.